# The Mechanical Properties and Damage Evolution of UHPC Reinforced with Glass Fibers and High-Performance Polypropylene Fibers

**DOI:** 10.3390/ma14092455

**Published:** 2021-05-09

**Authors:** Jiayuan He, Weizhen Chen, Boshan Zhang, Jiangjiang Yu, Hang Liu

**Affiliations:** 1Department of Bridge Engineering, Tongji University, Shanghai 200000, China; 1832271@tongji.edu.cn (J.H.); 86039@tongji.edu.cn (W.C.); yujj0904@tongji.edu.cn (J.Y.); 2Shandong Hi-Speed Company Limited, Jinan 250000, China; sdjjlh@163.com

**Keywords:** ultra-high-performance concrete, glass fibers, HPP fibers, concrete damage plasticity

## Abstract

Due to the sharp and corrosion-prone features of steel fibers, there is a demand for ultra-high-performance concrete (UHPC) reinforced with nonmetallic fibers. In this paper, glass fiber (GF) and the high-performance polypropylene (HPP) fiber were selected to prepare UHPC, and the effects of different fibers on the compressive, tensile and bending properties of UHPC were investigated, experimentally and numerically. Then, the damage evolution of UHPC was further studied numerically, adopting the concrete damaged plasticity (CDP) model. The difference between the simulation values and experimental values was within 5.0%, verifying the reliability of the numerical model. The results indicate that 2.0% fiber content in UHPC provides better mechanical properties. In addition, the glass fiber was more significant in strengthening the effect. Compared with HPP-UHPC, the compressive, tensile and flexural strength of GF-UHPC increased by about 20%, 30% and 40%, respectively. However, the flexural toughness indexes *I*_5_, *I*_10_ and *I*_20_ of HPP-UHPC were about 1.2, 2.0 and 3.8 times those of GF-UHPC, respectively, showing that the toughening effect of the HPP fiber is better.

## 1. Introduction

Ultra-high-performance concrete (UHPC) has been widely used in engineering structures because of its high strength, high toughness and high durability [1]. The incorporation of fibers can improve the ability to inhibit crack propagation, achieving the improved toughness of UHPC [2,3,4,5]. The reinforcing effect of steel fibers is better than that of nonmetallic fibers [6,7]. However, steel fibers exposed on the surface rust quickly and may hurt pedestrians [8]. Demand for nonmetallic fibers is significant in structures like landscape bridges and pedestrian bridges. Therefore, it is of great significance to further study the strengthening and toughening effects and mechanism of nonmetallic fibers on UHPC.

The mechanical properties of UHPC can be improved by adding nonmetallic fibers. Meng and Khayat [9] studied the fluidity, hydration reaction, autogenous shrinkage and pore structures of UHPC with carbon nanofibers and reported that the incorporation of carbon nanofibers reduced the porosity of UHPC by about 35%. Mehboob et al. [10] indicated that the polypropylene fiber could effectively improve the strength and ductility of UHPC. Yu et al. [11] found that the strain-hardening cementitious composite gained better crack control ability when mixing polyvinyl alcohol (PVA) fibers and polyethylene terephthalate (PET) fibers. Nonmetallic fibers can also improve the corrosion resistance of UHPC. Ghasemzadeh et al. [12] studied the microstructure of the PVA-reinforced UHPC, observing that PVA fibers reduced the diffusion and infiltration of chloride ions. Holubova et al. [13] studied the chemical durability of glass fibers (GF) in UHPC and observed no serious corrosion defects in glass fibers and low quantities of corrosion products in UHPC. Some studies indicated that the mechanical properties of the hybrid fiber-reinforced concrete were better than that of the single-doped fiber-reinforced concrete. Christ et al. [14] mixed steel fibers and polypropylene fibers into UHPC and claimed that the hybridization of different kinds of fibers increased the mixture’s toughness. Yoo et al. [15] applied four types of steel fibers in UHPC and found that the tensile performance of UHPC can be improved by the hybrid reinforcing system. Sorelli et al. [16] indicated that the performance of concrete mixed with different sizes of fibers could be effectively improved. Park et al. [17] claimed that small size fibers were beneficial to strain hardening, while large size fibers mainly determined the overall shape of the tensile stress–strain curve. In addition, mixing different elastic modulus fibers can also improve the integrity of concrete [18,19,20]. Thus, it is necessary to further study the reinforcing effects of different elastic modulus and different sizes of nonmetallic fibers on UHPC.

Among various nonmetallic fiber materials, glass fibers with small size and high elastic modulus can enhance UHPC [21,22,23]; high-performance polypropylene (HPP) fibers with large size and low elastic modulus can improve the toughness of concrete [24,25,26]. However, there is insufficient research on the effect of HPP fibers on UHPC. The research on the strengthening mechanism and mixing effect of glass fibers and HPP fibers is also insufficient. Thus, to improve the strengthening and toughening effects and mechanism of glass fibers and HPP fibers, the two kinds of fibers were selected to prepare UHPC by single doping or mixing in this research.

Although mechanical performance tests can intuitively reflect the effects of fiber reinforcement, numerical analysis is still necessary, considering the time consumption and material cost. Bahij et al. [27] studied the numerical simulation method of shear performance of UHPC beams and got a highly accurate result. Zhang et al. [28] established a discrete numerical simulation method to predict the fracture behavior of PVA-UHPC, and the experimental results agreed with the model well. Chen et al. [29] established the concrete damaged plasticity (CDP) model by ABAQUS to simulate the response of UHPC pi-girders, and the results verified the rationality of the model. Some studies also used the CDP model to simulate the three-dimensional failure of UHPC [30,31,32]. However, the existing studies pay little attention to the degradation process of UHPC during the loading process. By establishing the CDP model, this paper intends to analyze the damage evolution of specimens under force loading and study the mechanical behavior of UHPC.

In this paper, the mechanical properties of UHPC were tested by the axial compression test, the axial tension test and the 4-point bending test. With the experimental data, the mixing effect of glass fibers and HPP fibers was also quantitatively analyzed. Based on the analytical software ABAQUS, the CDP model of UHPC was established, and the simulation results were compared with the test results, helping analyze the damage evolution of UHPC. Moreover, this model could also provide a reference for the nonlinear finite element analysis of UHPC.

## 2. Experimental Study

### 2.1. Materials

This study tested and analyzed the properties of 9 UHPC proportions with various fiber contents. The properties of the fibers, including the glass fiber (GF) and the high-performance polypropylene (HPP) fiber, are shown in Table 1. Samples of the fibers are shown in Figure 1.

The UHPC matrix recipe is listed in Table 2, adopting the improved Andreasen and Andersen (A & A) particle packing model [33].

The raw materials included cement, mineral powders, aggregates, polycarboxylate superplasticizer and water. Portland cement P. II 52.5 was chosen. The mineral powders included the ultrafine silica fume with the particle size of 0.1–0.15 µm and the quartz powder with the particle size of 2.4 µm. The aggregates included the quartz sand I with the size of 0.15–0.2 mm and the quartz sand II with the size of 0.3–0.6 mm. The composition of the materials is shown in Table 3; the particle size distribution curves of the materials are given in Figure 2 [34].

The liquid polycarboxylate superplasticizer with a water-reducing rate of 36% was added. The denomination of the 9 groups of UHPC is given in Table 4.

During the preparation and casting process, the fibers dispersed uniformly in the slurry, and no fiber agglomeration was observed. After pouring, the specimens were placed in the room with a temperature of 20 ± 5 °C and relative humidity of more than 50% for 1 day. Then the specimens were removed and put into the curing room with the temperature of 20 ± 2 °C and the relative humidity of more than 95% for curing. After 28 days of curing, the specimens were tested.

### 2.2. Experimental Investigation

To obtain the compressive, tensile and bending properties of UHPC, the axial compression test was performed according to British standard BS EN 12390-3: 2009 [35]; the axial tensile test and the 4-point bending test were performed according to Chinese standard T/CBMF 37-2018 [36]. In the axial compression test, the cubic specimens of 100 mm × 100 mm × 100 mm were compressed axially with the loading rate of 0.6 MPa/s as seen in Figure 3a, and 6 specimens were tested for each UHPC proportion group. In the axial tensile test, the dog-bone specimens without stud were subjected to axial tension loads with the loading rate of 0.1 MPa/s, as seen in Figure 3b. Six specimens were tested for each UHPC proportion group. The middle part with the constant section (the shaded area in Figure 3b) is defined as the middle tensile zone, where the fracture is supposed to happen. Two strain gauges were pasted on two opposite sides in the middle tensile zone. In the 4-point bending test, the beam specimens of 100 mm × 100 mm × 400 mm (Figure 3c) were bent with the loading rate of 0.05–0.08 MPa/s before the initial crack and with the loading rate of 0.1 mm/min after the initial crack. Three displacement meters (Linear Variable Differential Transformer, abbreviated as LVDT) were arranged to measure the deflection. The loading was stopped when the load was about 10% lower than the peak value, and the specimens fractured in the pure bending zone were taken as effective specimens. Three specimens were tested for each group.

The average value was taken as the performance index of each proportion, eliminating the data with a difference of more than 15% from the average value. In the bending test, if the difference between the extreme value and the intermediate value was more than 15%, the intermediate value was taken as the performance index of the proportion. In the axial tensile test and the bending test, the stress–strain curve corresponding to the intermediate value of the measured strength was taken as the performance curve of each proportion.

The results of the axial compression tests, the axial tensile tests and the 4-point bending tests are shown in Table 5. The measuring uncertainty was ±1%. The bending toughness indexes *I*_5_, *I*_10_ and *I*_20_ are calculated according to ASTM C1018-97 [37].

## 3. Numerical Study

In this study, the UHPC damaged plastic model was established by the analytical software ABAQUS (6.14, 2014, Dassault Systemes Simulia Corp., RI, USA), and the loading process of the axial compression test, the axial tensile test and the 4-point bending test were simulated. The linear reduced eight-node hexahedral integral C3D8R solid element was adopted, and damage parameters were introduced to consider irreversible damage degradation of the material and simulate the inelastic mechanical behavior. The cubic models of 100 mm × 100 mm × 100 mm were established for the axial compression test, and the mesh division size was 5 mm. The prismatic models of 50 mm × 50 mm × 100 mm were established for the axial tensile test (corresponding to the shaded area in Figure 3b), and the mesh division size was 2.5 mm. The beam models of 100 mm × 100 mm × 400 mm were established for the 4-point bending test, and the mesh division size was 8 mm. To make the models have good convergence, the viscosity coefficient *μ* was taken as 0.005, and other concrete parameters were defined as follows [38,39].

### 3.1. Yield Function and Flow Rule

The CDP model defines the yield function of concrete material according to Equation (1),
(1)$$F\left(\bar{\sigma },{\tilde{\varepsilon }}^{pl}\right)=\frac{1}{1-\alpha }\left(\bar{q}-3\alpha \bar{p}+\beta \left({\tilde{\varepsilon }}^{pl}\right)\langle {\hat{\bar{\sigma }}}_{max}\rangle -\gamma \langle -{\hat{\bar{\sigma }}}_{max}\rangle \right)-{\bar{\sigma }}_{c}\left({\tilde{\varepsilon }}_{c}^{pl}\right)\leq 0$$
where: $\bar{\sigma }$ is the effective stress; ${\tilde{\varepsilon }}^{pl}$ is the hardening variables; is defined as 〈X〉=(|X|+X)/2; $\bar{q}$ is Mises equivalent effective stress; $\bar{p}$ is the effective hydrostatic stress; ${\hat{\bar{\sigma }}}_{max}$ is the maximum eigenvalue of $\bar{\sigma }$; ${\bar{\sigma }}_{c}$ and ${\bar{\sigma }}_{t}$ are the effective tensile and compressive cohesion stresses, respectively. Dimensionless material constants *α*, *γ* and function *β* are defined as:(2)$$\alpha =\frac{\left(\frac{f_{b0}}{f_{c0}}-1\right)}{\left(\frac{2f_{b0}}{f_{c0}}-1\right)},\beta \left({\tilde{\varepsilon }}^{pl}\right)=\frac{{\bar{\sigma }}_{c}\left({\tilde{\varepsilon }}_{c}^{pl}\right)}{{\bar{\sigma }}_{t}\left({\tilde{\varepsilon }}_{t}^{pl}\right)}\left(1-\alpha \right)-\left(1+\alpha \right),\gamma =\frac{3\left(1-K_{c}\right)}{2K_{c}-1}$$
where: *f_b_*_0_ is the initial equibiaxial compressive yield stress; *f_c_*_0_ is the initial uniaxial compressive yield stress; *K_c_* is the ratio of the second stress invariant on the tensile meridian to that on the compressive meridian. Referring to the existing literature and after model debugging, *f_b_*_0_/*f_c_*_0_ = 1.16 and *K_c_* = 0.667 were defined [40,41,42,43].

The CDP model adopts non-associated flow rule, and the plastic potential follows Drucker–Prager hyperbolic function:(3)$${\dot{\varepsilon }}^{pl}=\dot{\lambda }\frac{\partial G\left(\bar{\sigma }\right)}{\partial \bar{\sigma }}$$
(4)$$G=\sqrt{{\left(\in \sigma_{t0}\tan\varphi \right)}^{2}+{\bar{q}}^{2}}-\bar{p}\tan\varphi$$
where: φ is the dilation angle in the *p-q* plane, and φ=30° is defined [40]; $\sigma_{t0}$ is the uniaxial tensile stress at failure; ∈ is the eccentricity parameter, and ∈=0.1 is defined.

### 3.2. The Equivalent Plastic Strain and the Effective Stress

In the CDP model, the compressive and tensile strain are calculated according to Equations (5) and (6):(5)$$\{\begin{array}{l} \varepsilon_{c}={\tilde{\varepsilon }}_{c}^{in}+\varepsilon_{0c}^{el}={\tilde{\varepsilon }}_{c}^{pl}+\varepsilon_{c}^{el}\\ \varepsilon_{0c}^{el}=\frac{\sigma_{c}}{E_{0}},\varepsilon_{c}^{el}=\frac{\sigma_{c}}{\left(1-d_{c}\right)E_{0}} \end{array}$$
(6)$$\{\begin{array}{l} \varepsilon_{t}={\tilde{\varepsilon }}_{t}^{ck}+\varepsilon_{0t}^{el}={\tilde{\varepsilon }}_{t}^{pl}+\varepsilon_{t}^{el}\\ \varepsilon_{0t}^{el}=\frac{\sigma_{t}}{E_{0}},\varepsilon_{t}^{el}=\frac{\sigma_{t}}{\left(1-d_{t}\right)E_{0}} \end{array}$$
where: $\varepsilon_{c}$ and $\varepsilon_{t}$ are, respectively the compressive strain and the tensile strain; ${\tilde{\varepsilon }}_{c}^{in}$ and ${\tilde{\varepsilon }}_{t}^{ck}$ are, respectively the inelastic compressive strain and the inelastic tensile strain; $\varepsilon_{0c}^{el}$ and $\varepsilon_{0t}^{el}$ are, respectively the elastic compressive strain and the elastic tensile strain corresponding to initial stiffness of material; $\varepsilon_{c}^{el}$ and $\varepsilon_{t}^{el}$ are, respectively, the elastic compressive strain and the elastic tensile strain corresponding to actual stiffness of material after damage degradation; *d_c_* and *d_t_* are, respectively, the compressive damage parameter and the tension damage parameter, which reduce the material stiffness; *E*_0_ is the initial stiffness of the material.

The equivalent plastic strain ${\tilde{\varepsilon }}_{c\left(t\right)}^{pl}$, stress $\sigma_{c\left(t\right)}$ and effective stress ${\bar{\sigma }}_{c\left(t\right)}$ are calculated according to Equations (7)–(9) by introducing the damage parameter $d_{c\left(t\right)}$,
(7)$${\tilde{\varepsilon }}_{c\left(t\right)}^{pl}={\tilde{\varepsilon }}_{c}^{in}\left({\tilde{\varepsilon }}_{t}^{ck}\right)-\frac{d_{c\left(t\right)}}{1-d_{c\left(t\right)}}\cdot \frac{\sigma_{c\left(t\right)}}{E_{0}}$$
(8)$$\sigma_{c\left(t\right)}=\left(1-d_{c\left(t\right)}\right)E_{0}\left(\varepsilon_{c\left(t\right)}-{\tilde{\varepsilon }}_{c\left(t\right)}^{pl}\right)$$
(9)$${\bar{\sigma }}_{c\left(t\right)}=\frac{\sigma_{c\left(t\right)}}{1-d_{c\left(t\right)}}=E_{0}\left(\varepsilon_{c\left(t\right)}-{\tilde{\varepsilon }}_{c\left(t\right)}^{pl}\right)$$

### 3.3. Constitutive Relation and Damage Parameter

The stress–strain curve of UHPC under uniaxial compression adopted the constitutive model defined by Wu [44], as shown in Equation (10):(10)$$\{\begin{array}{cc} y=Ax+\left(6-5A\right)x^{5}+\left(4A-5\right)x^{6} & 0\leq x\leq 1\\ y=\frac{x}{\alpha {\left(x-1\right)}^{2}+x} & 1<x \end{array}{}_{}^{}\left\{ \begin{array}{l} x=\frac{\varepsilon_{c}}{\varepsilon_{0}}\\ y=\frac{\sigma_{c}}{f_{c}} \end{array}\right.$$
where: x and y are the dimensionless coordinates; $\varepsilon_{co}$ is the peak compressive strain; $f_{c}$ is the prismatic compressive strength.

The parameters in Equation (10) were determined by the fitting formulas [45], and $f_{cu}$ measured in axial compression test was used to define the constitutive relation. First, $f_{c}$ was obtained by Equation (11). Then *ε_c_*_0_, *E* and the dimensionless material constant *A* were calculated from Equations (12)–(14). *α* = 4.00 was defined [44], and other model parameters for each proportion are shown in Table 6. Moreover, the density ρ of UHPC was 2.5 × 10^3^ kg/m^3^; Poisson’s ratio ν was 0.2 in the model:(11)$$f_{c}=0.89f_{cu}$$
(12)$$\varepsilon_{co}=\left(6.7264f_{c}+2460.9\right)\times 10^{-6},80\;\mathrm{MPa}\leq f_{c}\leq 150\;\mathrm{MPa}$$
(13)$$E=\frac{10^{3}}{0.0172+\frac{0.8364}{f_{c}}},60\;\mathrm{MPa}\leq f_{c}\leq 220\;\mathrm{MPa}$$
(14)$$A=\frac{6.7264f_{c}+2460.9}{17.2f_{c}+836.4},80\;\mathrm{MPa}\leq f_{c}\leq 150\;\mathrm{MPa}$$

The rising section of axial tensile stress–strain curve of UHPC was approximately defined as a linear line, and the descending section adopted the exponential curve model proposed by Jiang [46], as shown in Equation (15):(15)$$\sigma =f_{t}e^{-\alpha_{t}\left(\varepsilon -\varepsilon_{cr}\right)}$$
where: $f_{t}$ is the ultimate tensile strength of concrete; $\varepsilon_{cr}$ is the strain at the peak value of concrete tensile stress; $\alpha_{t}$ is the softening coefficient, and the greater the $\alpha_{t}$ value is, the steeper the curve of descending section is. In this study, $f_{t}$ measured in the axial tensile test was used to define the constitutive relation, and $\alpha_{t}$= 1000 was defined.

According to the constitutive relation, the damage parameter *d* was calculated by area method based on the principle of energy loss, as shown in Equation (16):(16)$$d=1-\frac{A_{d}}{A_{0}},A_{d}=\int \sigma d\varepsilon ,A_{0}=\frac{1}{2}E_{0}\varepsilon$$
where: $A_{d}$ is the strain energy of damaged material, which represents the area under the stress–strain curve. $A_{0}$ is the strain energy of non-damaged material, which represents the triangle area under a straight line with the slope $E_{0}$.

The analysis process of the CDP model is shown in Figure 4.

## 4. Results and Discussions

### 4.1. Compressive Properties

#### 4.1.1. Comparison of Test and Numerical Values

Figure 5 shows the axial compression test results and the simulated values, and the difference is within 5%, indicating that the cubic numerical model is reasonable and reliable. For GF-UHPC, the difference of $f_{cu}$ between G1 and G2 is only 0.8%, while that between G2 and G3 is 15.4%, reaching 131.7 MPa. For HPP-UHPC, $f_{cu}$ of H3 is the highest, which is 6.9% and 13.8% higher than that of H1 and H2, respectively. For G/H-UHPC (UHPC mixed with glass fibers and high-performance polypropylene fibers), $f_{cu}$ of GH1, GH2 and GH3 are about 100 MPa, and $f_{cu}$ increases slightly with the increase of glass fiber content. With the same fiber content, $f_{cu}$ of G1, G2 and G3 are 13.6%, 20.0% and 21.7% higher than that of H1, H2 and H3, respectively, showing that glass fibers have a more efficient reinforcement effect because of the high-strength and high-elastic modulus. Moreover, the compressive performance of G3 (the UHPC with 2.0% fiber content) is the best.

Figure 6 shows the typical damage evolution and failure mode under UHPC cubes under compression, taking G1 as an example. The numerical model is cut to facilitate observation of the internal damage by the damage cloud diagram (DAMAGEC) in Figure 6a. When the displacement d of the model is loaded to 0.30 mm (ε=0.003), the material degradation first occurs in the area near the UHPC cube corners and the diagonal of surface; when d = 0.45 mm, it develops into a double “X” distribution (the white dotted line in Figure 6). As d continues to increase, the damaged area expands, and the stiffness degradation degree of the diagonal and middle area becomes higher. Finally, it develops into an “X” type, which is consistent with the pyramid-shaped or hourglass-shaped failure bodies formed in the axial compression test (Figure 6b).

The experimental and the numerical results show that the binding effect of fibers on the UHPC matrix is naturally limited when the fiber content is low. The negative effect of uneven fiber dispersion would be more prominent at the low fiber content. As shown in Figure 7, due to the uneven dispersion of fibers during stirring, there is “region I” where fiber content is much lower than the total content in cement paste. As the number of fibers in “region I” is very low, it is difficult to bridge cracks effectively, which forms the weak part and affects the strength of UHPC specimens. On the contrary, in the case of high fiber content, even if there are also regions with lower fiber numbers, the high total fiber content makes fibers in those regions still bridge cracks effectively, forming the “region II” in Figure 7. Moreover, in other regions with higher fiber number, densely and randomly distributed fibers form a three-dimensional network structure, which could be closely linked to the UHPC matrix and prevent the development and penetration of microcracks, thereby significantly improving the compressive strength, such as the “region III” in Figure 7.

As shown in Figure 8, glass fibers and UHPC matrix are jointly subjected to friction through contact surface before UHPC cracking; after UHPC cracking, glass fibers through cracks continue to share external force and slow down the expansion of microcracks so as to achieve reinforcement effect. Compared with glass fibers, the diameter of HPP fibers is larger (0.8–1.5 mm), and the length is longer (30 mm). The number of fibers in a certain area is less, resulting in the significantly weaker ability to inhibit microcracks. Once the microcracks are further expanded and penetrated into macrocracks, HPP fibers could reinforce and toughen the matrix (Figure 9).

Figure 10 shows the crushed cubic specimens with different proportions. It could be observed that the broken blocks were connected as a whole by fibers, and the integrity of specimens from high to low is G/H-UHPC > HPP-UHPC > GF-UHPC. Moreover, both microcracks and macrocracks in G/H-UHPC could be restrained, which greatly reduces the degree of disintegration and spalling of UHPC.

#### 4.1.2. Mixing Effect of Fibers

To quantitatively analyze the mixing effect of glass fibers and HPP fibers, and the mixing effect coefficient was calculated according to Equation (17) [47],
(17)$$\{\begin{array}{c} R=\frac{\eta -\left(\eta_{1}\lambda_{1}+\eta_{2}\lambda_{2}\right)}{\eta_{1}\lambda_{1}+\eta_{2}\lambda_{2}}\\ \lambda_{1}+\lambda_{2}=1,\lambda_{1}=\frac{V_{1}}{V},\lambda_{2}=\frac{V_{2}}{V} \end{array}$$
where: *R* is the mixing effect coefficient of fibers, comparing the experimental value of G/H-UHPC with the linear superposition of that of GF-UHPC and HPP-UHPC; η is the performance parameter of G/H-UHPC; $\eta_{1}$ and $\eta_{2}$ represent the performance parameters of GF-UHPC and HPP-UHPC, respectively; $\lambda_{1}$ and $\lambda_{2}$ represent the volume proportion of glass fibers and HPP fibers in G/H-UHPC, respectively; $V_{1}$ and $V_{2}$ are the volume of glass fibers and HPP fibers in G/H-UHPC, respectively; V is the total volume of fibers in G/H-UHPC.

Figure 11 shows the relationship between $f_{cu}$ and glass fibers contents (line chart), as well as the relationship between the mixing effect coefficient and $f_{cu}$ (histogram). The mixing effect of $f_{cu}$ is negative, as all the mixing effect coefficients are negative. The number of glass fibers is low in G/H-UHPC, which leads to the formation of “region I” but not “region III” in the UHPC matrix (Figure 7). As a result, $f_{cu}$ of G/H-UHPC is lower than that of GF-UHPC and is closer to that of HPP-UHPC.

### 4.2. Tensile Properties

#### 4.2.1. Comparison of Test and Numerical Values

Figure 12 shows the test values and the simulation values of *f_t_*, and the data deviation is within 5%. Figure 13 shows the stress–strain curves of the axial tensile test and the CDP model, and the simulation curves are close to the test curves, indicating that the numerical model is reasonable and reliable. The constitutive relation selected could be used for finite element analysis of UHPC without tensile strain hardening behavior.

According to Figure 12 and Table 5, as for GF-UHPC, $f_{t}$ is above 7 MPa and *E* is about 45 GPa; $f_{t}$ increases by 3.1% and 1.4% when fiber content increases by 0.5% from 1.0% to 2.0%. As for HPP-UHPC, $f_{t}$ is 5–6 MPa and *E* is 37–44 GPa; $f_{t}$ increases by 7.7% and 3.9% when fiber content increases by 0.5% from 1.0% to 2.0%, and the amplification is larger than that of GF-UHPC. As for G/H-UHPC, $f_{t}$ is above 5 MPa and *E* is above 40 GPa; and $f_{t}$ increases with the increase of glass fiber content. At the same fiber content, $f_{t}$ of G1, G2 and G3 are 38.5%, 32.6% and 29.3% higher than that of H1, H2 and H3, respectively, indicating that glass fibers are more effective in strengthening tensile properties than HPP fibers.

According to the stress–strain curves are shown in Figure 13, the tensile failure process first goes through the elastic stage, in which the load increases linearly. Then, brittle failure occurs after cracking, and there is no strain strengthening. *E* and *ε_cr_* of GF-UHPC are slightly higher than those of HPP-UHPC, while those of G/H-UHPC in between. The length of the glass fiber is shorter, and the anchorage friction force is smaller. As a result, fibers would be pulled out quickly when cracks widening, making the specimen brittle. As for HPP fibers, the size is larger, and the number of fibers is less under a certain volume content, leading to a weak constraint effect of cracks and brittleness.

Taking G1 as an example, the typical von Mises stress cloud diagram of the axial tensile model is shown in Figure 14. When the displacement d = 0.163 mm, the load reaches the peak value, and it is difficult for glass fibers to continue bridging microcracks, then the load decreases with microcracks expanding rapidly. When d = 0.168 mm, most of the fibers have been pulled out, which could not prevent the extension of macrocracks, and the load decreases to 13.3% of the peak value. When d = 0.020 mm, the load drops to 1.0 kN and the model loses its bearing capacity.

The presentative fracture sections are shown in Figure 15. The maximum pore diameter of G3 is 3.8 mm, and the number and size of pores of G3 are significantly higher than those of G1 and G2, while those of H3 are also slightly higher than those of H1 and H2. With the increase of fiber content, more bubbles may be easily generated during the mixing and pouring process of UHPC, which could have a negative impact on the mechanical properties. In addition, it could also be observed in Figure 15 that the failure of glass fibers is mainly due to interface sliding pull-out, while there are more fractured HPP fibers. Thus, the matrix bonding and anchoring ability of large-size fibers are stronger, which is conducive to give full play to the strength of HPP fibers.

#### 4.2.2. Mixing Effect of Fibers

The mixing effect coefficients were calculated according to Equation (17). The relationship between *f_t_*, *E*, *ε_cr_* and glass fibers contents (line chart) and the relationship between the mixing coefficient and *f_t_*, *E* and *ε_cr_* (histogram) are obtained, as shown in Figure 16, Figure 17 and Figure 18. The mixing effect of *f_t_* is close to zero effect, and the mixing effects of *E* and *ε_cr_* are weak positive effects, indicating that the tensile properties of UHPC could be strengthened by mixing glass fibers and HPP fibers (like “1 + 1 > 2”). However, the positive effect is weak, and the maximum enhancement range is 7.72% (GH1 in Figure 17).

### 4.3. Bending Properties

#### 4.3.1. Comparison of Test and Numerical Values

Figure 19 shows the test values and simulation values of *f_f_*, and the data deviation is within 5%. Figure 20 shows the experimental and simulated bending load-deflection curves. As the simulation of *f_f_* is consistent, the model could be considered valid in the elastic stage and the descending stage. However, the simulated curves have no strain hardening behavior because of the brittleness of tensile curves. Although the model cannot show the toughening effect of HPP fibers, it is still beneficial to analyze the stress distribution and the crack propagation process in the linear elastic stage.

According to Figure 19 and Table 5, *f_f_* of GF-UHPC and HPP-UHPC decrease slightly when the fiber content increases from 1.0% to 2.0%. This may be caused by excessive bubbles. With the same fiber content, G1, G2 and G3 are 21.5%, 21.6% and 20.1% higher than H1, H2 and H3 in initial crack strength, and 46.3%, 35.7% and 25.0% higher in flexural strength. Moreover, *f_cr_* and *f_f_* of G/H-UHPC increase with the increase of glass fibers content, indicating that glass fibers have a better effect on strengthening flexural properties of UHPC than HPP fibers.

According to Figure 20, the loading process of GF-UHPC specimens could be divided into two stages. Stage I is the elastic stage. The load before initial cracking increases linearly, and glass fibers bear force cooperating with the UHPC matrix because of its high elastic modulus. In stage II, the load declines to brittle failure rapidly after initial cracking, and then it is difficult for small-size glass fibers to bridge macrocracks, making glass fibers continuously and rapidly pulled out.

The loading process of HPP-UHPC specimens could be divided into four stages: the linear elastic stage, the load declining stage, the strain hardening stage and the strain-softening stage. In stage I, unlike glass fibers, the reinforcement effect of HPP fibers is low because of the low elastic modulus, and it is the UHPC matrix that mainly bears external force. In stage II, the load decreases rapidly after initial cracking; H1, H2 and H3 decline to 22.57%, 26.15% and 36.19% of the peak load, respectively. Moreover, the decrease range decreases with the increase of HPP fiber content. In stage III: it could be observed that a wide main crack occurs in the pure bending section of the specimen. Most of the UHPC in the tensile zone is out of work, and HPP fibers continue to bear external force so that the load increases nonlinearly. In stage IV: the specimen was obviously deformed, and the UHPC withdrawing from work expends with the crack propagation. Near the main crack, some HPP fibers are pulled out or broken, then the load declines slowly.

The loading process of G/H-UHPC could be divided into four stages like HPP-UHPC, but GH2 and GH3 have no obvious strain hardening stage. In stage I, the higher the glass fibers content is, the higher the elastic limit load is, as glass fibers bear force cooperating with the UHPC matrix. After initial cracking, the load decreases rapidly like GF-UHPC and HPP-UHPC in stage II, then the long HPP fibers bridge macrocracks, making the load of GH1 with high HPP fibers content rise again while GH2 and GH3 with low HPP fibers content enter the strain-softening stage directly in stage III. As for GH1, with the main crack continuing to expand and widen, HPP fibers are pulled out or broken one after another so that the load declines slowly in stage IV.

According to Figure 20 and Table 5, the bending toughness indexes increase with the increase of the high HPP fibers content, and the flexural toughness of H3 is the highest. At the same volume content, the bending toughness indexes $I_{5}$ of H1, H2 and H3 are 1.21, 1.21 and 1.28 times of that of G1, G2 and G3; $I_{10}$ of H1, H2 and H3 are 1.89, 2.05 and 1.99 times of that of G1, G2 and G3; $I_{20}$ of H1, H2 and H3 are 3.63, 3.99 and 3.88 times that of G1, G2 and G3, respectively, indicating that the toughening effect of HPP fibers is much higher than that of glass fibers.

Figure 21 shows the von Mises stress and tensile damage cloud diagram of the bending model. As the cloud diagrams of other proportions are similar to G1, so no longer repeated list. When the deflection $d_{f}$ = 0.21 mm, the load reaches the peak value, and the model is about to crack as material at the bottom of the beam degenerates. When $d_{f}$ = 0.26 mm, there is the strip-shaped irrecoverable tensile damage in the tensile zone, and the load declines rapidly to 37.9% of the peak value with the model cracking. When $d_{f}$ = 0.39 mm, as the degenerated area continues to expand, the compression zone of UHPC decreases, and the load declines to 15.7% of the peak value. When $d_{f}$ = 1.44 mm, the degenerated area extends to the whole height of the beam, and the load declines to less than 3 kN, showing that the bearing capacity has been lost. The failure mode of the CDP model is basically consistent with the experimental situation, reflecting the process of brittle failure. The cloud diagram of UHPC tensile damage is also corresponding to cracks of the beam specimen.

When the load declined to 10% of the peak load, the maximum crack width *W**_crmax_* was measured, as shown in Figure 22. All the cracks are located in a pure bending area, and only one main crack occurs when the specimen is fractured, with no other microcracks observed near the main crack. *W**_crmax_* of GF-UHPC, HPP-UHPC and G/F-UHPC are 0.8–3.5 mm, 9.0–18.6 mm and 7.6–13.6 mm, respectively (two values with significant deviation are excluded), showing that short glass fibers cannot keep bridging cracks after initial cracking, but long HPP fibers could further prevent the expansion of macrocracks.

#### 4.3.2. Mixing Effect of Fibers

Calculate the mixing effect coefficients according to Equation (17). The relationship between *f_cr_*, *f_f_*, *I*_5_, *I*_10_, *I*_20_ and glass fibers proportion (line chart) and *R* of *f_cr_*, *f_f_*, *I*_5_, *I*_10_ and *I*_20_ (histogram) are shown in Figure 23, Figure 24 and Figure 25. The mixing effects of *f_cr_*, *f_f_*, *I*_5_, *I*_10_ and *I*_20_ are positive when glass fibers content is high and zero or negative when HPP fibers content is high, indicating that only when there are more glass fibers can the flexural toughness of UHPC be strengthened like “1 + 1 > 2”.

## 5. Conclusions

In this paper, the mechanical properties of nine groups of UHPC proportions with different fiber types and contents were tested, and the damage evolution of the specimens was numerically analyzed by ABAQUS, revealing the strengthening and toughening effects and mechanism of glass fibers and HPP fibers on UHPC. Based on the above results and discussions, the following conclusions could be drawn:The mechanical properties of GF-UHPC and HPP-UHPC with 2.0% fiber content were better in the research. With the increase of fiber content, the amplification of mechanical properties of HPP-UHPC was slightly higher than that of GF-UHPC. Moreover, it should be pointed out that the mechanical properties of UHPC did not necessarily increase with the growing fiber content in the research;Glass fibers had a more efficient reinforcement effect than HPP fibers. With the same fiber volume content, the cubic compressive strength, the tensile strength and the flexural strength of GF-UHPC were about 20%, 30% and 20–50% higher than those of HPP-UHPC, as glass fibers had higher elastic modulus and stronger inhibition of microcracks;The toughening effect of HPP fibers was better than that of glass fibers. With the same fiber volume content, the bending toughness indexes *I*_5_, *I*_10_ and *I*_20_ of HPP-UHPC were about 1.2 times, 2.0 times, and 3.8 times of those of GF-UHPC, as long HPP fibers could further play a role in the connection of macrocracks after initial cracking;The strength indexes of G/H-UHPC increased with the increase of glass fiber content, and the toughness indexes increased with an increase of HPP fiber content. The mixing effects of the tensile strength, the flexural strength, the bending toughness indexes *I*_5_, *I*_10_ and *I*_20_ at high glass fibers content were positive, as more “region I” (Figure 7) exiting in G/H-UHPC with low glass fibers amount and it was difficult to obtain a positive effect like “1 + 1 > 2”;The material degradation area of the CDP model of UHPC axial compression was approximately the “X” shape, which was consistent with the pyramid-shaped or hourglass-shaped failure bodies formed in the test;According to the fitting formulas summarized in reference [45], the peak compressive strain, the elastic modulus and the dimensionless material constant *A* could be calculated from the cubic compressive strength of UHPC and used in numerical simulation. The simulation results were in good agreement with the experimental results, which proves that the exponential constitutive relation could be applied to finite element analysis of UHPC without tensile strain hardening, and the model establishment method was reliable.

## Figures and Tables

**Figure 1 materials-14-02455-f001:**
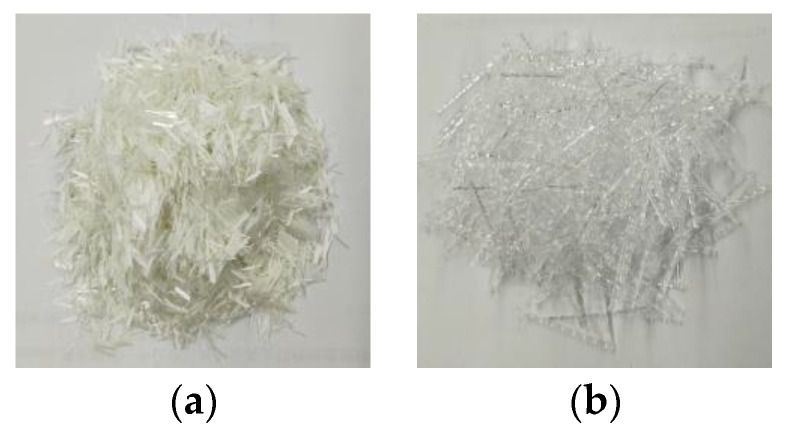
Samples of the fibers: (**a**) glass fibers (GF); and (**b**) high-performance polypropylene (HPP) fibers.

**Figure 2 materials-14-02455-f002:**
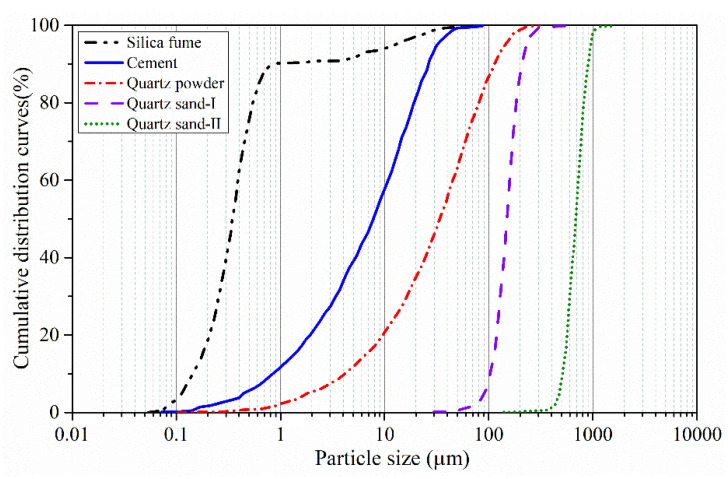
The particle size distribution curves of the materials [34].

**Figure 3 materials-14-02455-f003:**
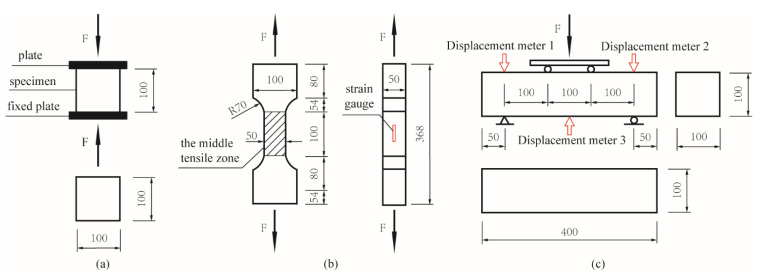
Test setup (units: mm): (**a**) The axial compression test; (**b**) the axial tensile test; (**c**) the 4-point bending test.

**Figure 4 materials-14-02455-f004:**
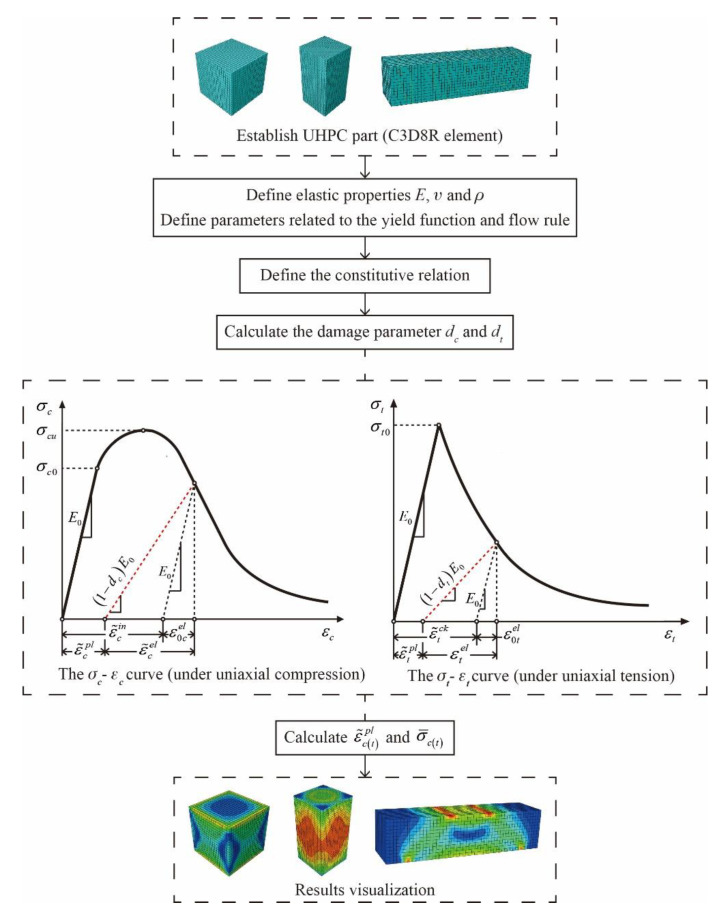
The analysis process of the numerical model. UHPC—ultra-high-performance concrete.

**Figure 5 materials-14-02455-f005:**
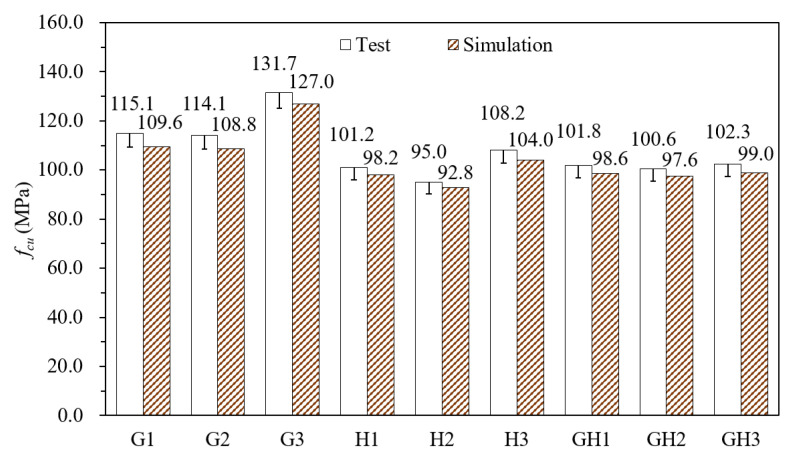
Test values and simulation values of *f_cu_* (cubic compressive strength).

**Figure 6 materials-14-02455-f006:**
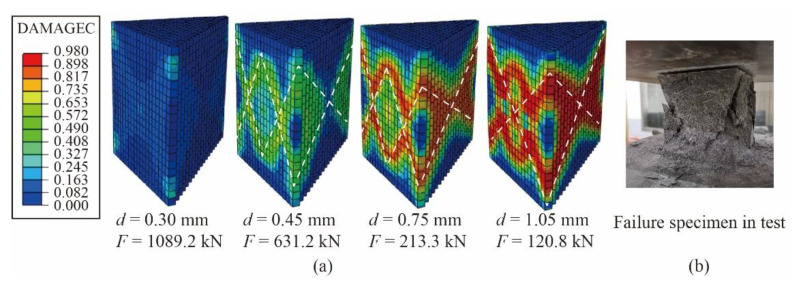
Damage under compression: (**a**) compression damage cloud diagram; and (**b**) failure specimen of UHPC cube (G1).

**Figure 7 materials-14-02455-f007:**
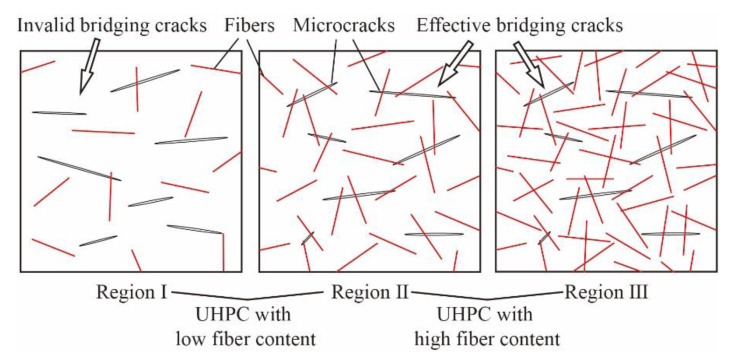
Effects of fiber content on bridging cracks.

**Figure 8 materials-14-02455-f008:**
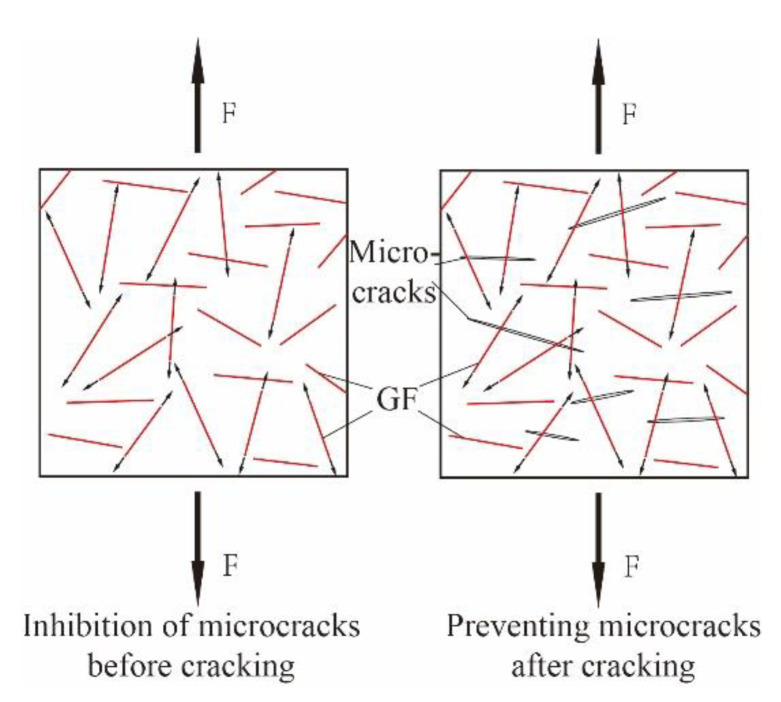
Reinforced mechanism of (GF).

**Figure 9 materials-14-02455-f009:**
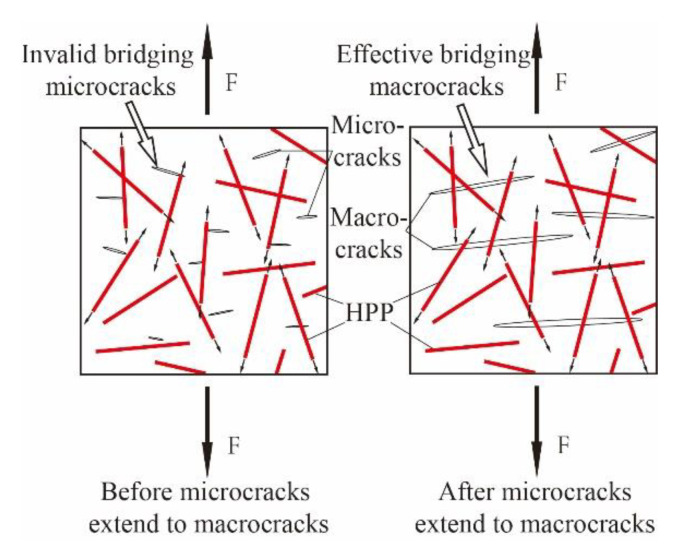
Reinforced mechanism of (HPP).

**Figure 10 materials-14-02455-f010:**
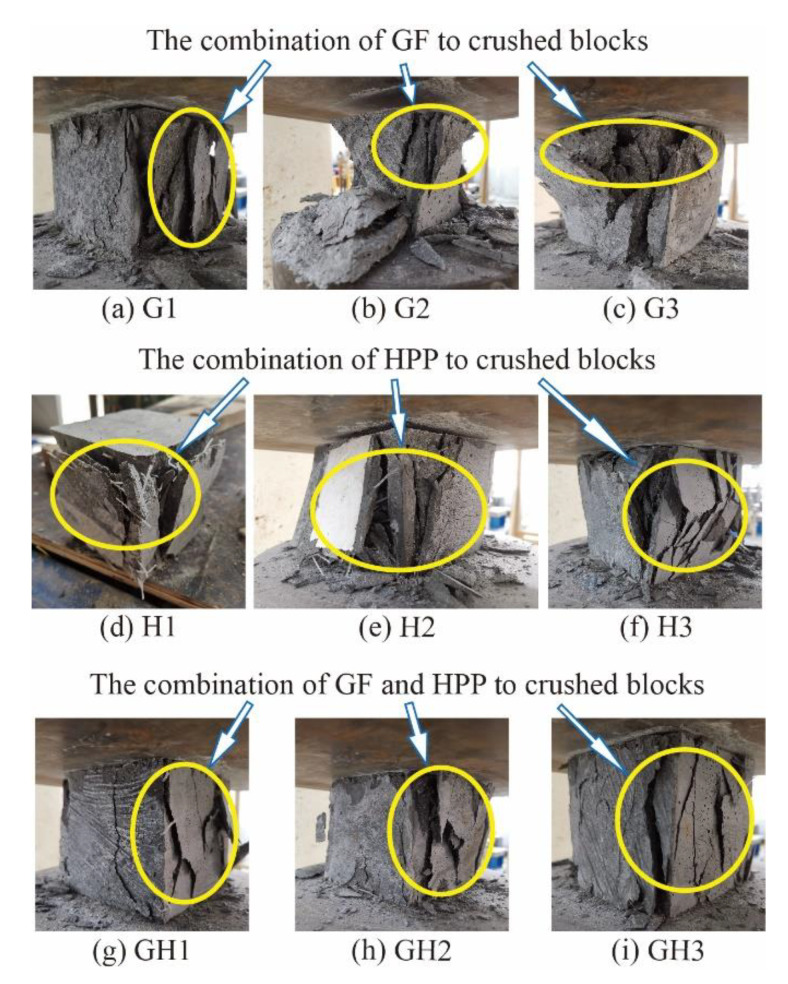
Crushed cube specimens.

**Figure 11 materials-14-02455-f011:**
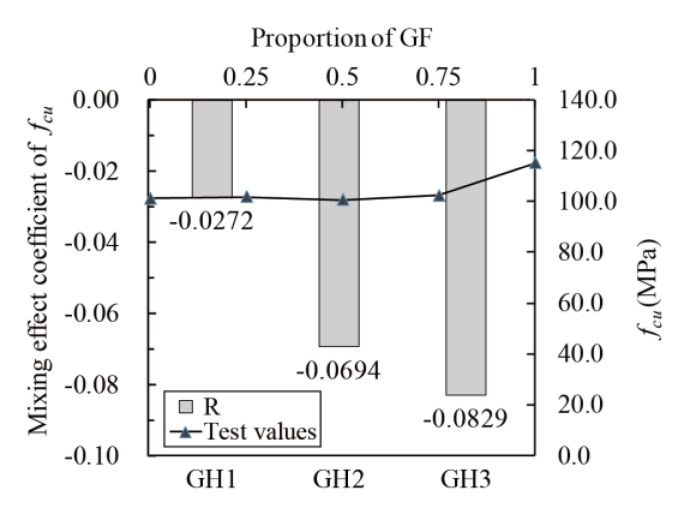
Mixing effect coefficient of *f_cu_*.

**Figure 12 materials-14-02455-f012:**
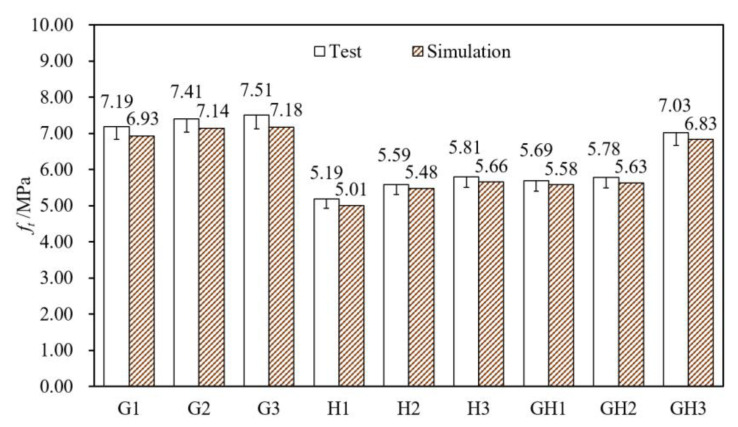
Test values and simulation values of *f_t_* (tensile strength).

**Figure 13 materials-14-02455-f013:**
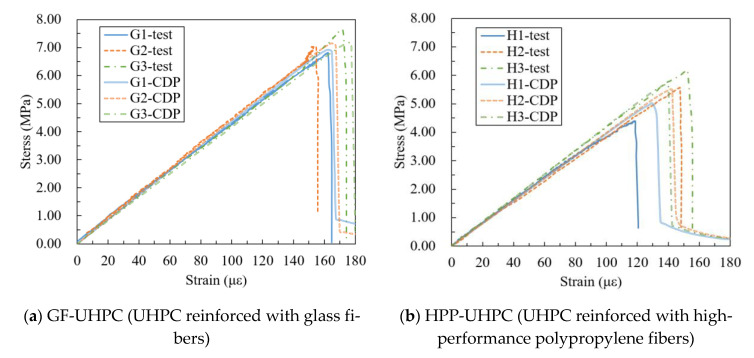
Experimental and simulated tensile stress–strain curves.

**Figure 14 materials-14-02455-f014:**
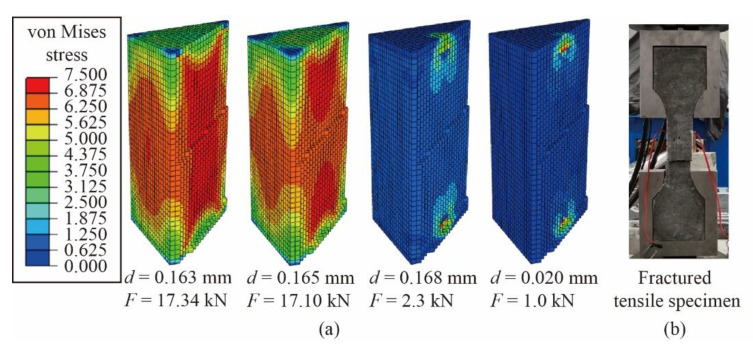
(**a**) Von Mises stress (MPa) cloud diagram of the axial tensile model and (**b**) fractured specimen (G1).

**Figure 15 materials-14-02455-f015:**
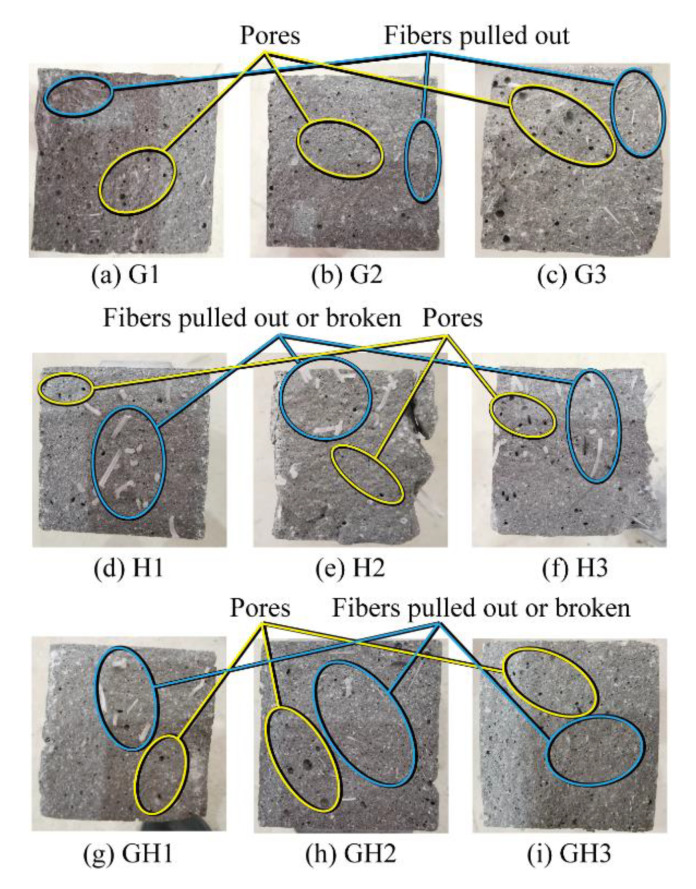
Fracture sections of dog-bone specimens.

**Figure 16 materials-14-02455-f016:**
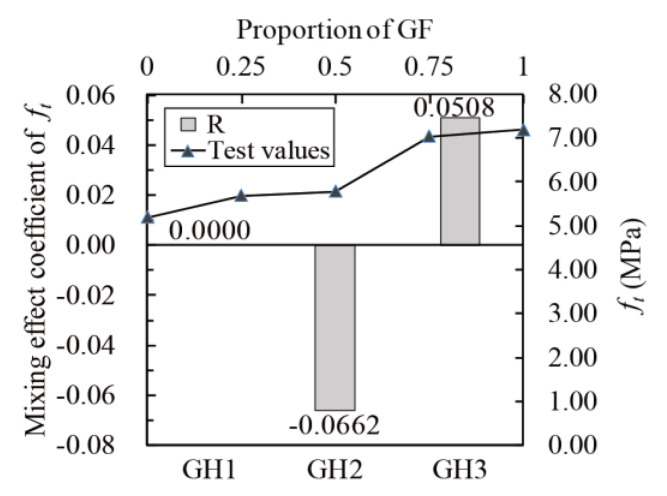
Mixing effect coefficient of *f_t_*.

**Figure 17 materials-14-02455-f017:**
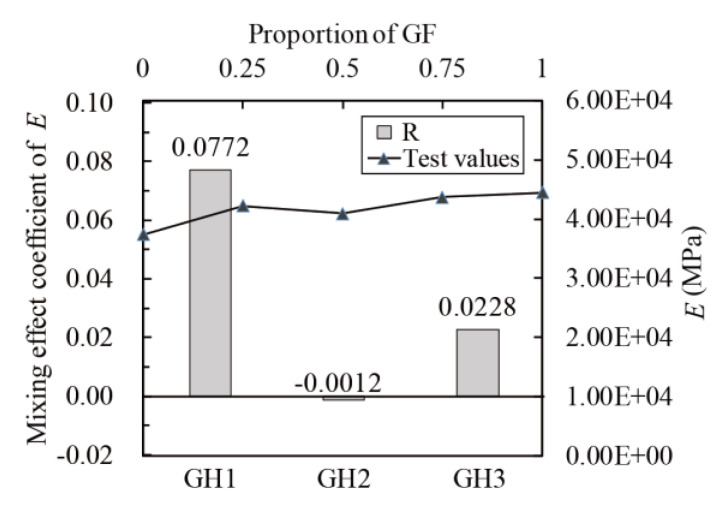
Mixing effect coefficient of *E* (Tensile modulus of elasticity).

**Figure 18 materials-14-02455-f018:**
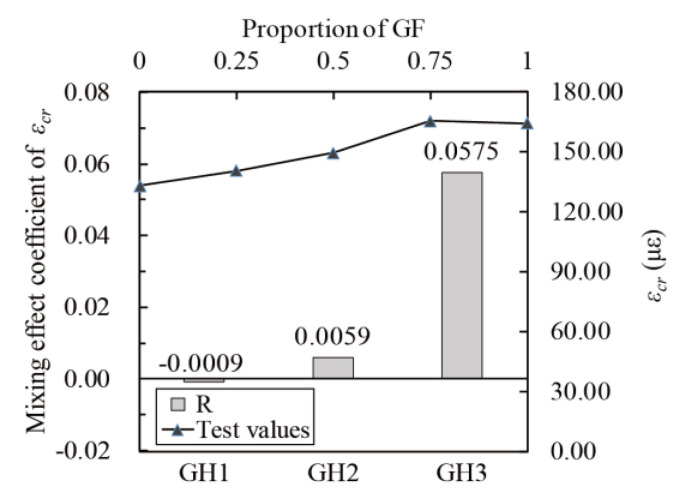
Mixing effect coefficient of *ε_cr_* (peak tensile strain).

**Figure 19 materials-14-02455-f019:**
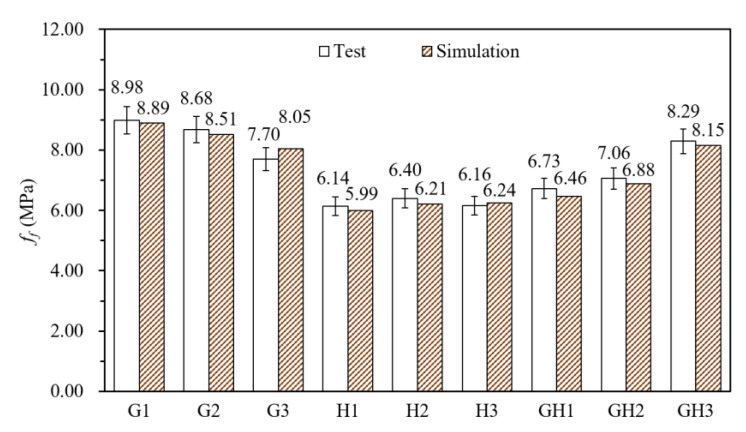
Test values and simulation values of *f_f_*.

**Figure 20 materials-14-02455-f020:**
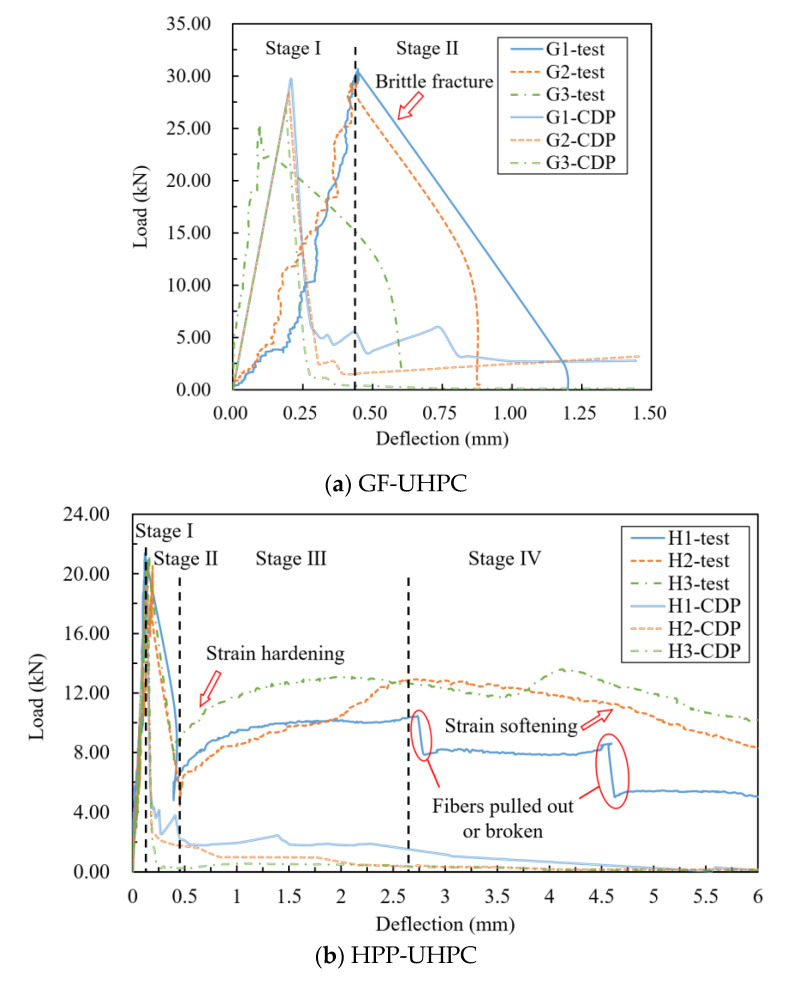
Experimental and simulated bending load-deflection curves.

**Figure 21 materials-14-02455-f021:**
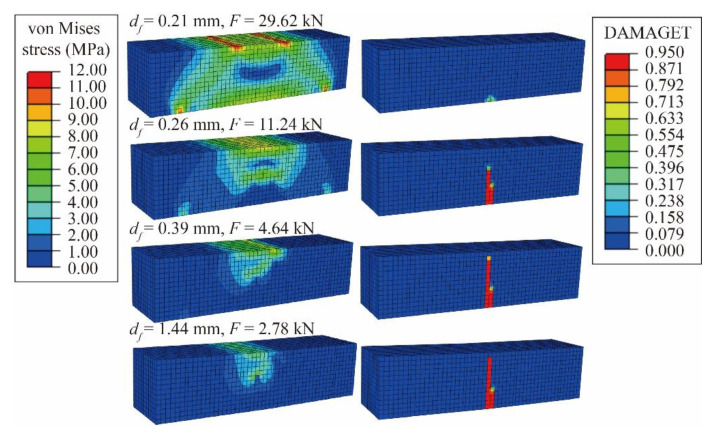
Von Mises stress (MPa) and tensile damage cloud diagram of the 4-point bending test.

**Figure 22 materials-14-02455-f022:**
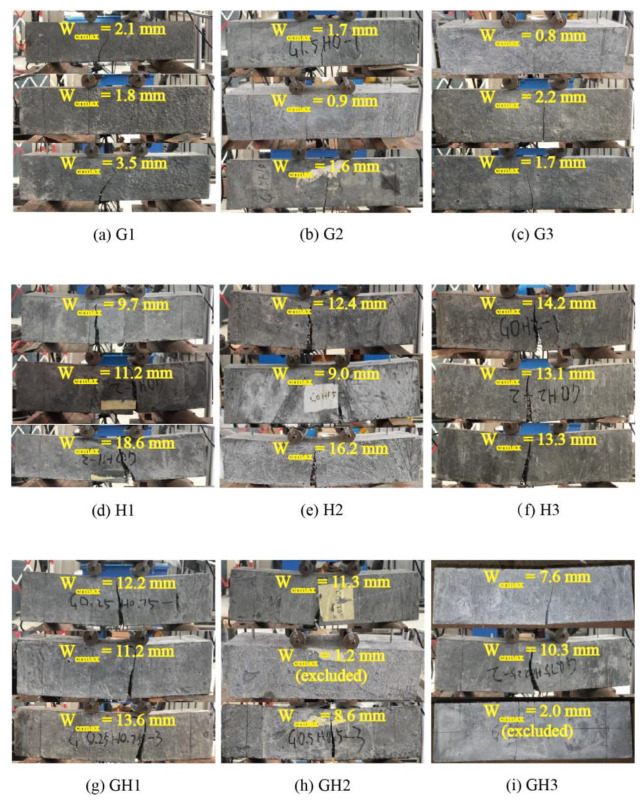
Fractured specimens of the 4-point bending test.

**Figure 23 materials-14-02455-f023:**
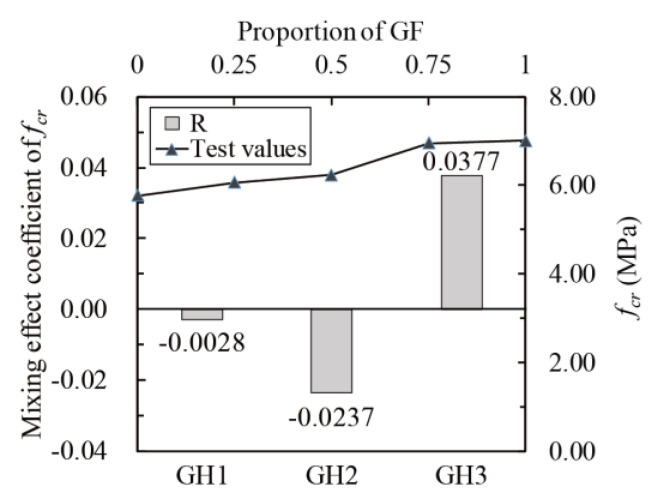
Mixing effect coefficient of *f_cr_*.

**Figure 24 materials-14-02455-f024:**
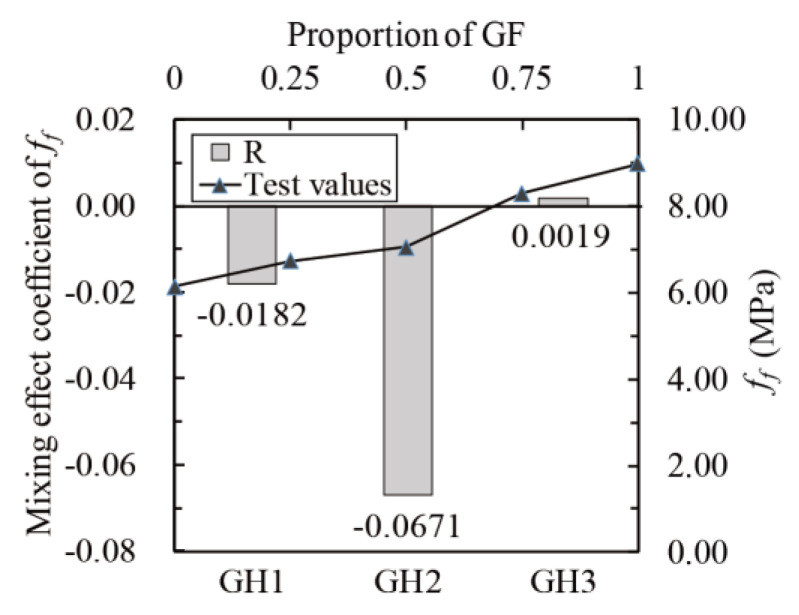
Mixing effect coefficient of *f_f_* (flexural strength).

**Figure 25 materials-14-02455-f025:**
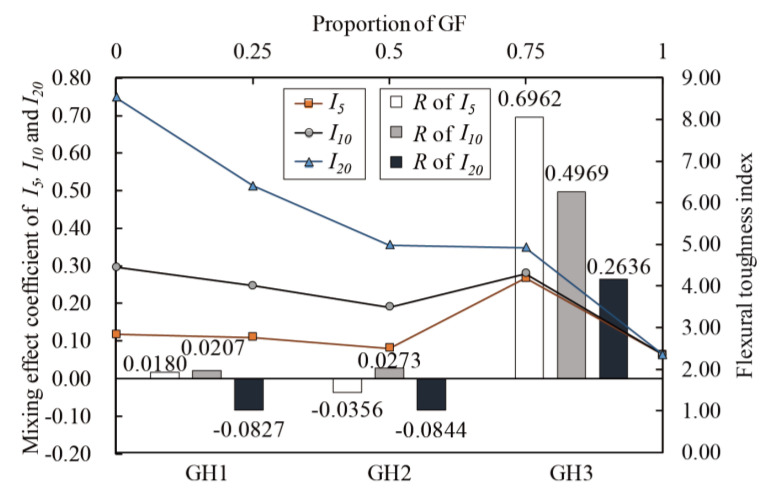
Mixing effect coefficient of *I*_5_, *I*_10_ and *I*_20_.

**Table 1 materials-14-02455-t001:** Properties of the fibers.

Types	Length (mm)	Diameter (mm)	Tensile Strength (MPa)	Elastic Modulus (MPa)	Density (kg/m^3^)
GF	6	0.015	1500	71,000	2540
HPP	30	0.8–1.5	500	5000	910

GF—glass fiber, HPP—high-performance polypropylene.

**Table 2 materials-14-02455-t002:** UHPC (ultra-high-performance concrete) benchmark mix (kg/m^3^).

Cement	Silica Fume	Quartz Powder	Superplasticizer	Water	Quartz Sand I	Quartz Sand II
750	90	263	12	191	306	714

The water-to-cement ratio is 0.255.

**Table 3 materials-14-02455-t003:** Composition of the materials (mass fraction, %) [34].

Composition	CaO	SiO_2_	Fe_2_O_3_	Al_2_O_3_	SO_3_	K_2_O	Na_2_O	LOI
Cement	67.87	20.25	4.01	2.76	2.61	0.49	0.21	1.80
Silica fume	0.25	98.00	0.10	0.20	0.52	0.25	0.20	0.48
Quartz powder	0.31	97.40	0.30	0.31	0.65	0.37	0.26	0.40
Quartz sand	0.25	91.78	0.88	4.34	0.02	2.06	0.22	0.45

The cement is provided by Jiangnan Onoda Cement Co., Ltd. (Nanjing, China).

**Table 4 materials-14-02455-t004:** Volume contents of the fibers (%).

Types	G1	G2	G3	H1	H2	H3	GH1	GH2	GH3
GF	1	1.5	2	0	0	0	0.25	0.5	0.75
HPP	0	0	0	1	1.5	2	0.75	0.5	0.25

G means the glass fiber, while H means the high-performance polypropylene (HPP) fiber.

**Table 5 materials-14-02455-t005:** Test results.

Name	Axial Compression Test	Axial Tensile Test			Four-Point Bending Test				
	*f_cu_* (MPa)	*f_t_* (MPa)	*E* (MPa)	*ε_cr_* (με)	*f_cr_* (MPa)	*f_f_* (MPa)	*I* _5_	*I* _10_	*I* _20_
G1	115.1	7.19	4.45 × 10^4^	164.15	7.01	8.98	2.35	2.35	2.35
G2	114.1	7.41	4.42 × 10^4^	166.81	6.69	8.68	2.32	2.32	2.32
G3	131.7	7.51	4.58 × 10^4^	180.41	6.77	7.70	2.67	2.67	2.67
H1	101.2	5.19	3.74 × 10^4^	132.79	5.77	6.14	2.84	4.45	8.53
H2	95.1	5.59	4.09 × 10^4^	140.72	5.50	6.08	2.81	4.76	9.25
H3	108.2	5.81	4.43 × 10^4^	138.52	5.63	6.16	3.41	5.30	10.35
GH1	101.8	5.69	4.22 × 10^4^	140.50	6.06	6.73	2.76	4.0	6.41
GH2	100.6	5.78	4.09 × 10^4^	149.35	6.24	7.06	2.50	3.49	4.98
GH3	102.3	7.03	4.37 × 10^4^	165.30	6.95	8.29	4.19	4.30	4.92

*f_cu_*: cubic compressive strength, *f_t_*: tensile strength, *E*: tensile modulus of elasticity, *ε_cr_*: peak tensile strain, *f_cr_*: initial crack strength, *f_f_*: flexural strength.

**Table 6 materials-14-02455-t006:** Compression constitutive parameters of CDP (concrete damaged plasticity) model.

Group	*f_cu_* (MPa)	*f_c_* (MPa)	*ε_c_* _0_	*E* (MPa)	*A*
G1	115.05	102.39	0.0031496	39,419.11	1.21
G2	114.14	101.58	0.0031442	39,318.17	1.22
G3	131.65	117.17	0.0032490	41,087.27	1.14
H1	101.15	90.02	0.0030664	37,748.80	1.29
H2	94.97	84.52	0.0030294	36,906.50	1.32
H3	108.24	96.33	0.0031089	38,636.40	1.25
GH1	101.78	90.58	0.0030702	37,830.93	1.28
GH2	100.60	89.53	0.0030631	37,676.56	1.29
GH3	102.34	91.08	0.0030736	37,903.38	1.28

## Data Availability

Data is contained within the article.

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
