# Peer review of "The Mechanical Properties and Damage Evolution of UHPC Reinforced with Glass Fibers and High-Performance Polypropylene Fibers"

_materials, 2021, doi:10.3390/ma14092455_

Round 1

Reviewer 1 Report

In the Reviewer opinion the research paper entitled “The strengthening and toughening effects and mechanism of glass fibers and high performance polypropylene fibers on UHPC” is good.

Some comments which greatly enhance the understanding of the paper and its value are presented below. Specific issues that require further consideration are:

  1. The title of the manuscript is matched to its content.
  2. In the Reviewer’s opinion, the current state of knowledge relating to the manuscript topic has been presented, but the author's contribution and novelty are not enough emphasized.
  3. In the Reviewer’s opinion, the bibliography, comprising 37 references, is rather representative.
  4. Please improve the quality of drawings.
  5. An analysis of the manuscript content and the References shows that the manuscript under review constitutes a summary of the Author(s) achievements in the field. However, the introduction needs more attention.
  6. Conclusion needs to be more revised and extended.
  7. In the Reviewer’s opinion the manuscript should be published in the journal after minor revision.

Reviewer 2 Report

Ultra-high performance concretes are being developed for some decades, initially with metallic fibers and, for some years, with plastic fibers. However their use is still relatively limited which can be partly explained by the difficulty to guarantee given performances because of the intrinsic difficulty to formulate such concrete and the lack of numerical tools to asses their properties.

The present manuscript exposes an investigation of the strengthening and toughening effects and mechanism of glass fibers and high performance polypropylene fibers on UHPC. The manuscript is composed of 5 parts: an introduction section, an ‘Experimental study’ section, a ‘Numerical study’ section, a ‘Results and discussions’ and a conclusive section. Experimental mechanical results (compressive tests, tensile tests and bending tests) are compared to numerical results obtained using the concrete damage plasticity implemented in Abaqus. Such a comparison may be interesting to the readers of the journal but important clarifications must be made before an eventual publication, especially regarding the experimental tests and the results analysis.

Please find the comments hereafter:

- l 86: please detail the composition of the cement and provide a reference to the manufacturer

- l 87 – 89: if available the particle size distribution curves of the materials should be given. If it is not possible particle sizes in micrometers should be added

- l 91: mixing and preparation of UHPC are of paramount importance. Please provide a reference to the standard used

- Table 1: this table giving the denomination of the samples might be given after Table 3 presenting the mix proportions

- Table 3: please specify the water-to-cement ratio

- l 100: curing condition, age at testing must be given

- l 109: precise it is 4-point bending

- l 132: Is CDP model freely available in Abaqus? Please give again a reference (Guan et al. 23?)

- l 147: von Mises should be used instead of Mises throughout the document

- l 177: the fitting procedure of the model should be clarified. Is the model fitted only with the compressive test measurements?

- l 208: please precise the 2% fiber content relates to G3

- l 208: the authors explain the detrimental effect of fibers due to the potential increase of porosity. Porosity measurements must be provided in my opinion.

- figure 3: is it possible (and useful) to add experimental compressive curves?

- l 258: the physical meaning of the mixing effect should be presented here (before line 322)

- l 389 and figure 18: the authors should discuss for which stages the CDP model can be considered as valid. Does it depend on the fibers type? I am personally not convince by the results of the model for bending curves

Reviewer 3 Report

The submitted paper is aimed at the evaluation of the strengthening and toughening effect of glass fibers and high performance polypropylene fibers on UHPC. Basically, the paper is interesting, well written and organized, but some improvements and corrections must be made before its acceptation for publication.

  1. In section 1. Introduction, the novelty of the presented research must be clearly stated as many papers have been already published in the studied field of concrete research.
  2. Section 2.1 Materials – the unit of the particle size of quartz powder and quartz sand is missing. Please, complete.
  • Section 2.2 – it is not clear, how the sample size was chosen in the conducted mechanical tests. Usually, the compressive strength is measured on 150 mm cubes and the bending strength test is realized on 400/150/150 mm prisms. Please, explain and comments.
  1. Section 2.2 – was there any standard followed in the mechanical tests?
  2. Table 4 – information on measuring uncertainty must be introduced. Please, complete.
  3. Table 3 – please, use superscript in unit kg/m3.
  • Conclusions – points 1 and 2 – the authors relate the mechanical performance and stiffness of the examined materials to the changes in porosity which is expected to be changed with the dosage of GF ad HPP fibers. It seems to be correct, however any information on the porosity of the examined UHPC must be provided. At least total porosity of the developed concrete must be introduced. Moreover, the information on pore size distribution might be interesting for the detailed characterization of the fibers’ effect. In this case, MIP analysis could be used.
  • Information on the dynamic elastic modulus and bulk density of the tested UHPC might be presented.

As the above listed comments and suggestions will be well addressed in the revised version of the paper, I will be able to reconsider my opinion and recommend paper for publication.

Round 2

Reviewer 2 Report

The authors corrected the manuscript based on the various reviewers comments. Most of the reviewers remarks have been addressed and the precision of the article, especially regarding the materials and methods description, has been improved.

Only porosity measurements have not been provided by the authors. Porosity increase with fiber content was legitimately suspected to decrease mechanical properties in the first manuscript version. However, as it is not the main focus of the article and without sufficient experimental evidence, the authors decided to remove this explanation. It is an acceptable option and the manuscript might be considered for publication in my opinion.

Reviewer 3 Report

Thank you for revision of the paper. I recommend it for publiction.